# Social Integration, Social Support, and All-Cause, Cardiovascular Disease and Cause-Specific Mortality: A Prospective Cohort Study

**DOI:** 10.3390/ijerph16091498

**Published:** 2019-04-27

**Authors:** Jinke Tan, Yafeng Wang

**Affiliations:** 1School of Political Science and Public Administration, East China University of Political Science and Law, Shanghai 201620, China; tanjinke@ecupl.edu.cn; 2Department of Epidemiology and Biostatistics, School of Health Sciences, Wuhan University, Wuhan 430071, China

**Keywords:** social integration, social support, mortality, cardiovascular disease, cohort

## Abstract

Social relationships are associated with all-cause mortality. Substantial uncertainties remain, however, for the associations of social relationships with mortality from subtypes of cardiovascular disease (CVD) and major non-vascular diseases. This prospective cohort study estimated mortality risks according to social support and social integration utilizing a nationally representative sample of 29,179 adults ages 18 years and older. Cox proportional hazards regression models were employed. Social integration, but not social support was associated with all-cause mortality risk. For CVD mortality, social integration predicted a 33% lower risk (HR = 0.67, 95% CI = 0.53–0.86). The results were similar in magnitude for heart disease mortality. Participants with the highest social integration level had a 53%, 30%, and 47% decreased mortality risk of diabetes, Alzheimer’s disease, and chronic lower respiratory diseases (CLRD) than those with the lowest level. These social integration associations were linear and consistent across baseline age, sex and socioeconomic status. We did not observe an association of social integration with the risk of cancer mortality. Our findings support the linear association of social integration but not social support with mortality from a range of major chronic diseases in the US adult population, independent of socioeconomic status (SES), behavioral risk factors, and health status.

## 1. Introduction

People with greater social connectedness and stronger ratings of social support tend to live longer [1]. In 1988, House and colleagues demonstrated a causal relationship between social relations and the risk of mortality [2]. Recently, many epidemiological and clinical studies have established social relationships as one of several key domains relevant to mortality risk [3,4,5]. A meta-analysis including 148 studies showed that both functional and structural relationships were inversely associated with a 50% reduction in mortality rates, which was comparable to common health risk factors such as smoking and diabetes [3].

Although the associations between low social connections and increased the likelihood of survival has been repeatedly shown, the biological, behavioral, or social pathways that may drive these relationships are not well elucidated. Social relationships have been hypothesized to improve health through various mechanisms, such as stress buffering, role modeling and social control of health behaviors [6]. Those with poor social relationships have less healthy behaviors, lower levels of immunity and increased inflammation which may lead to bad health outcomes [7]. Numerous previous studies have evaluated the link between social relations and all-cause, cardiovascular disease (CVD), and cancer mortality, but few studies have investigated the associations between heart disease mortality, stroke mortality, and other mortality of chronic diseases such as diabetes, Alzheimer’s disease, and chronic lower respiratory diseases (CLRD). Furthermore, apart from socioeconomic status (SES), such as income and employment status, these associations are modified by smoking, obesity, diabetes and hypertension. In addition, the protective effect was found to be inconsistent in structural and functional aspects of social relations. In addition, whether the form of the association between social relationships and cause-specific mortality is linear or has a threshold effect remains unclear.

We conducted a prospective cohort study using a nationally-representative sample of U.S. adults who participated in the 2001 US National Health Interview Survey (NHIS) to assess the independent association between social integration and the risks of all-cause, CVD, cancer, and other cause-specific mortality in a 15-year follow-up. In addition, we also assessed the association between social support and mortality. Subsequently, this study also evaluated whether the form of the social relation-mortality association is linear and whether the association was consistent by age, sex, and socioeconomic status.

## 2. Materials and Methods

### 2.1. Study Sample

NHIS is an ongoing national cross-sectional survey, conducted annually by the National Center for Health Statistics in collaboration with the US census bureau since 1957, and that uses a multistage sample design to monitor the health of the US civilian non-institutionalized population. Data from the 2001 NHIS were employed (*n* = 33,326, aged 18–85+ years). Data from adult participants in this wave were linked to death corticated data from the National Death Index (NDI) through 31 December 2015 (*n* = 31,379). Participants who were lost to follow-up were censored on the last confirmed date of presence in the study area. After excluding participants with missing data, 29,179 participants comprised the final sample.

### 2.2. Assessment of Mortality Events and Follow-Up

In 2001, 33,326 sample adults completed the NHIS. The survey records were matched to the NDI and subsequent vital status ascertainment through 31 December 2015. Person-years of follow-up were calculated for each participant from the data of the starting point to the date of death or end of the study period (31 December 2015). Underlying causes of death were classified according to the 10th revision of the International Statistical Classification of Diseases, Injuries, and Causes of Death (ICD-10). The endpoints included all-cause and cause-specific deaths from cancer (C00 to C97), CVD (I00-I78), heart disease (I00-I09, I11, I13, I20-I51), stroke (I60 to I69), CLRD (J40-J47), diabetes (E10-E14) and Alzheimer’s disease (G30).

### 2.3. Primary Exposure

Participants’ social support was assessed by the question “how often do you get the social and emotional support you need? Always, usually, sometimes, rarely, never.” The rarely and never categories were combined for analysis due to small frequencies. Rarely/never, sometimes, usually, always are scored on a scale from 1 to 4. The composite social integration was created and based on eight binary questions [5]. Four questions assessed the past two weeks contacts with relatives or friends, either in person or over the telephone. Three questions assessed attending a religious service, a group social activity, or going out to eat in the past two weeks. The final social integration item was marital status, defined as whether participants were married/living with a partner or not. For each question, scored 0 was no and 1 was yes. The social integration score ranged from 0 to 8. Owing to low frequencies across subgroups (age, sex, education level, income level and employment status) and low frequencies across number of cause-specific deaths, the social integration score was unavailable to analysis separately. To produce more stable estimates, social integration scores were collapsed into four categories of social integration indexes which reflected 0–3, 4–5, 6 and 7–8 social contacts [8].

### 2.4. Covariates

Demographic variables included age, sex, race (non-Hispanic white, non-Hispanic black, Hispanic, other non-Hispanic), education level (less than high school degree, high school degree or equivalent and more than high school degree), income level (family income to poverty ratio (PIR) ≤ 1, 1 < PIR ≤ 4, PIR ≥ 4), and employment status (never worked, working, retired, or out of work). Lifestyle variables were from self-reported responses for these questionnaires. Lifestyle variables included body mass index (BMI), smoking status, alcohol intake, history of hypertension (no versus yes), history of diabetes (no versus yes), history of heart disease (no versus yes), history of stroke (no versus yes), and history of cancer (no versus yes). BMI was calculated as weight in kilograms divided by height squared and was categorized as <25, 25–30 and >30 kg/m^2^. Respondents were defined as never smokers if they had smoked less than 100 cigarettes in their lifetime. Former smokers are defined as those who had smoked at least 100 cigarettes in their lifetime but did not currently smoke. Participants were classified as current smokers who reported having smoked more than 100 cigarettes and currently smoked. Respondents were categorized into three alcohol consumption groups: (1) lifetime abstainers: <12 drinks in one’s lifetime; (2) former drinkers: ≥12 drinks in a previous year; (3) current drinker: drinking now and ≥12 drinks in one’s lifetime.

### 2.5. Statistical Analysis

Baseline characteristics of study participants were reported by using percent-ages for categorical variables. In addition, we tested for differences between the categories of social integration and social support among participant characteristics by using chi-square test. We used a multilevel Cox proportional hazards model to calculate the hazard ratios (HRs) and 95% confidence intervals (CIs) of social integration and social support for all-cause and cause-specific mortality. Furthermore, we used total social integration scores (0–8 scores) and a social integration index as continuous variables to assess whether there was a linear dose relationship, respectively. In addition, we performed a subgroup analysis according to age groups, gender, and levels of socioeconomic status. Several sensitivity analyses were also conducted to evaluate the robustness of the results: (1) exclusion of individuals who died within two years after the interview; (2) exclusion of participants with chronic diseases including heart disease, stroke and cancer at baseline; (3) limiting analyses to CVD or cancer patients. Accounting for the complex survey design employed in NHIS, all analyses were conducted using the final weights which represent a product of weights for corresponding units computing in each of the sampling stage, recommended by the Centers for Disease Control and Prevention. The Taylor series method was used to account for sample weights, primary sampling and clustering units. The analyses were all conducted using Stata13.0 statistical software (Stata Corp LP, College Station, TX, USA) and all *p*-values refer to two-tailed tests. Statistical significance was set at *p* < 0.05.

## 3. Results

### 3.1. Baseline Characteristics

Demographics of the overall study population and subpopulations by social integration index were demonstrated in Table 1. Among the 29,179 participants included, 55.8% was less than 70 years old; 51.5% were women and 73.9% were non-Hispanic white. Compared to a low social integration index (0–3 scores), participants with a high social integration index (7–8 scores) were more likely to be females, ≤45 years old, non-Hispanic white ethnicity, employed, to have more education, more income, more social support, not to smoke, to be alcohol-users, and to present no history of hypertension, diabetes, heart disease and stroke. The mean follow-up was 13.8 years. During 0.4 million person-years of follow-up, a total of 5071 deaths were recorded, including 1231 from cancer, 1141 from CVD (889 and 252 deaths attributed to heart disease and stroke, respectively), 255 from CLRD, 175 from diabetes, and 128 from Alzheimer’s disease.

### 3.2. Associations of Social Integration with All-Cause, and Specific-Cause Mortality

We used Cox proportional hazards regression to evaluate the associations between social integration and all-cause, and specific-cause mortality. Compared with the least level of social integration, the fully adjusted HRs for all-cause mortality ranged from 0.94 to 0.70 with increasing levels of social integration which showed that higher level of social integration was associated with a decrease risk of mortality (Table 2). For CVD mortality, compared to those in the lowest social integration group, those in the highest social integration group had a 33% reduced risk of CVD mortality (HR = 0.67, 95% CI = 0.53–0.86). In the fully adjusted models, the results were similar in magnitude for heart disease mortality (HR = 0.65, 95% CI = 0.49–0.85). Social integration was associated with about 18% lower mortality from stroke, although the confidence interval of the HR was wide due to the relatively small number of deaths (*n* = 252). For diabetes mortality, subjects with the highest social integration level conferred a HR of 0.47 (95% CI = 0.26–0.84), compared with those with lowest social integration level after fully adjusting the covariates. What’s more, the fully adjusted HR reduced by approximately 50% when the index of social integration increased from just Level I to Level II. Social integration was also inversely associated with the risk of chronic lower respiratory diseases (HR = 0.53, 95% CI = 0.31–0.88) and Alzheimer’s disease (HR = 0.70, 95% CI = 0.37–1.32), respectively. However, we did not observe a significant association between social integration and a decreased risk of cancer mortality (HR = 0.94, 95% CI = 0.74–1.19). For each outcome above, there was an approximately log-linear dose–response relationship with the social integration score or index (all *p* for trend < 0.01; Table 3).

### 3.3. Associations of Social Support with All-Cause, and Specific-Cause Mortality

In the fully-adjusted model, we observed that the social support was not associated with all-cause, and specific-cause mortality risk (all-cause mortality: HR = 0.99, 95% CI = 0.96–1.02; CVD mortality: HR = 0.99, 95% CI = 0.93–1.06; Cancer mortality: HR = 1.00, 95% CI = 0.93–1.08; AD mortality: HR = 1.07, 95% CI = 0.89–1.28; CLRD: HR = 1.00, 95% CI = 0.89–1.13; Diabetes disease mortality: HR = 1.02, 95% CI = 0.88–1.19; All *p* values >0.1; Table 4).

### 3.4. Subgroup Group Analysis and Sensitivity Analyses

The results of our subgroup analysis showed that social integration was inversely associated with survival in different age, gender, socioeconomic status groups (Table 5). However, the magnitude of the impact of social integration from the lowest level (0–3) to the second level (4–5) is slightly larger in participants which were males, ≤65 years old, with moderate education, high income, and to have never worked. After excluding only deaths that occurred during the two years after baseline, there were similar results of analyses for the association of social integration with all-cause, CVD and cancer mortality (Table 6). Exclusion of participants with chronic diseases included heart disease, stroke and cancer at baseline did not materially change the observed associations above. The associations of similar directions and magnitudes were also observed when limiting analyses to CVD or cancer patients. To see whether the mortality hazards were exclusively associated with marital status, we disaggregated social integration into marital status and the remainder of the measured items, and marital status removed from social integration and entered separately did not obviously alter the relationship of other variables to mortality risk (data not shown).

## 4. Discussion

This prospective study found that social integration but not social support was robustly associated with mortality. As structural social relationships, social integration was associated with about a 30% lower risk of overall mortality. The social integration association was linear and consistent across age groups, sex, income, education and employment status. In the fully adjusted model, participants with highest social integration level had a 33–53% decreased mortality risk of CVD, especially heart disease, and other mortality of chronic diseases such as diabetes, Alzheimer’s disease, and CLRD than those with the least level. However, social integration was not associated with cancer mortality risk.

Social relationships consist of structural and functional aspects. However, whether functional and structural dimensions are equally important for mortality remain unclear. Some studies showed that structural social relationships, as a consequence of social participation itself, but not supportive functions, could improve physical health [9], and the supportive functions of social relationships may not represent the dimension of social relations inversely associated with mortality. Recently, Barger et al. showed that this association of functional support with all-cause mortality was not significant after SES adjustment [5]. In agreement with these results, our study did not observe a significant independent relationship between social support and all-cause mortality, which is also consistent for other cause-specific mortality. In our study, social integration or relationship quantity, which reflects participation in a broad range of social relationships, was demonstrated to be associated with all-cause mortality and other cause-specific mortality from chronic diseases apart from cancer [10,11].

In our study, the linear pattern of the link between social integration and cause-specific mortality was consistent with that between social integration and all-cause mortality no matter when using a summary social integration variable or social integration index, which provided further support the conclusion that people with moderate to high levels of social integration are at lower mortality risk [3,5].

We found a strong association between social integration and CVD mortality. Previous studies showed an association between high levels of social integration and improved health-promoting behaviors. Those with higher social integration were more likely to take more and regular physical activity to maintain health during leisure time, were less likely to smoke or more likely to quit smoking, and had better adherence to a healthy lifestyle and compliance to medical regimens [12,13,14]. A lack of social integration could result in negative psychological states such as anxiety or depression [15]. These adverse health behaviors and mood would lead to the development of coronary heart disease and stroke and increase morbidity and mortality risk [16]. Apart these aspects, social integration was related to chronic low-grade inflammation with lower levels of inflammatory cytokines include interleukin (IL)-1, tumor necrosis factor (TNF), and C-reactive protein [17], which appears critical in the progression of cardiovascular disorders. IL-1 and TNF-a can induce the binding of low-density lipoprotein (LDL) to vessel walls and the deposition of blood lipids, which causes atherosclerosis at early stages of cardiovascular disease [18]. In addition, long-term inflammation has been related to a wide range of chronic diseases, such as diabetes [19]. In our study, we found that the association was stronger with the risk of diabetes mortality than that of other chronic disease mortality, which could be partly explained by the stronger relationship between behavioral risk factors (e.g., smoking, obesity, smoking and physical activity) and the incidence risk of diabetes [20].

To date, previous studies evaluating the link between social integration and the risk of Alzheimer’s disease mortality are scarce. The present study showed that social integration may cause a 30% reduction in mortality rates of Alzheimer’s disease, although the small number of deaths (*n* = 128) led to a wide confidence interval of the HR estimate. The previous cohort study demonstrated that social integration had a protective effect against cognitive decline and dementia, which could be the main reason for the association between social integration and increased risk of Alzheimer’s disease mortality [21].

The strengths of the present study are its prospective cohort design and the use of a nationally representative adult sample. Our study also adjusted several potential confounders, especially related to the development of mortality, such as smoking, hypertension and diabetes. In addition, this study comprehensively investigated the associations for all-cause, all-cancer, CVD, and other cause-specific mortality, which are potentially more informative in understanding the mechanisms for the association between social integration and mortality. There are also some limitations to this study. Firstly, the measurements of social integration and social support were merely a one-time assessment at baseline, and covariates were also measured at baseline. Although social integration has a strong test–retest reliability over the course of follow up, the present study did not account for possible changes of social integration and other variables during the analysis period. Secondly, a single item was used to assess social support, which is potentially less reliable, although the previous study showed that this support item used has excellent predictive validity [22]. Finally, the longitudinal study is still observational, which may represent reverse or bi-directional causality. Mendelian randomization using gene variants as an instrumental variable could be a good study design to avoid confounding bias and infer causality in the future.

## 5. Conclusions

In conclusion, our findings suggest that social integration, but not social support, was robustly associated with mortality from a range of major chronic diseases in linear patterns in the US population, independent of SES, behavioral risk factors, and health status. To promote social integration, community interventions were included as a part of national social and health policies. Future studies should evaluate the extent by which behavioral interventions improve survival in the general population.

## Figures and Tables

**Table 1 ijerph-16-01498-t001:** Baseline characteristics based on social integration index in 2001 US National Health Interview Survey Participants With 15-year Vital Status Ascertainment (*n* = 29,179).

Variables	Social Integration Index					*p* Value
	Overall	I (0–3 Scores)	II (4–5 Scores)	III (6 Scores)	IV (7–8 Scores)	
	*(n* = 29,179)	(*n* = 2706)	(*n* = 7445)	(*n* = 7170)	(*n* = 11,858)	
Age, years						<0.001
18–45	55.8	46.8	53.6	59.4	56.7	
45–65	29.3	31.8	28.6	27.5	30.3	
65–	14.8	21.4	17.8	13.1	13.1	
Sex						<0.001
Female	51.5	47.8	48.5	51.9	53.4	
Race						<0.001
Hispanic	10.6	14.4	10.8	9.1	10.6	
Non-Hispanic White	73.9	66.7	72.4	76.3	74.8	
Non-Hispanic Black	11.3	14.5	12.5	10.0	10.8	
Non-Hispanic Other	4.2	4.5	4.3	4.6	3.9	
Education level						<0.001
Less than high school degree	17.4	35.4	23.5	15.5	11.9	
High school degree	29.3	33.1	32.1	29.5	27.0	
More than high school degree	53.3	31.5	44.4	55.0	61.0	
Income						<0.001
Low	11.1	22.8	15.0	10.3	7.3	
Moderate	49.1	55.1	53.8	49.2	45.6	
High	39.8	22.1	31.2	40.5	47.1	
Social support						<0.001
Never/rarely	5.4	17.7	8.2	3.9	2.6	
Sometimes	12.3	22.9	17.0	13.2	7.4	
Usually	34.3	28.5	34.5	37.1	33.8	
Always	48.0	31.0	40.4	45.8	56.2	
Employment status						<0.001
Employed	67.1	51.8	62.8	70.0	70.5	
Retired	14.1	18.1	15.7	12.6	13.4	
Not currently working	14.9	23.7	17.2	14.1	12.4	
Has never worked	3.9	6.6	4.3	3.3	3.6	
BMI (kg/m^2^)						<0.001
<25	41.7	41.3	40.6	43.5	41.2	
25–30	35.5	32.7	35.3	34.1	36.9	
>30	22.9	26.1	24.0	22.4	21.9	
Smoking status						<0.001
Never cigarette	54.6	42.6	47.6	52.3	61.7	
Former cigarette	22.4	21.3	22.0	23.1	22.5	
Current cigarette	23.0	36.2	30.4	24.6	15.8	
Alcohol intake						<0.001
Lifetime abstainer	21.7	26.3	20.3	17.8	23.8	
Former drinker	14.7	22.9	17.3	13.5	12.5	
Current drinker	63.6	50.8	62.4	68.7	63.8	
Physician-diagnosed disease						
Hypertension	23.7	31.7	27.9	22.0	20.9	<0.001
Diabetes	6.4	10.3	7.7	5.7	5.3	<0.001
Heart disease	7.6	10.0	9.1	6.9	6.8	<0.001
Stroke	2.3	5.0	3.0	2.0	1.6	<0.001
Cancer	7.0	7.6	7.9	6.7	6.7	<0.001

**Table 2 ijerph-16-01498-t002:** Hazard ratios for all-cause, cardiovascular, and cause-specific mortality by received social integration index.

Social Integration Index	HR	Lower	Upper	*p* Value
All-cause mortality				
0–3	1	Ref.		
4–5	0.94	0.84	1.04	0.23
6	0.77	0.68	0.87	<0.001
7–8	0.70	0.63	0.79	<0.001
Cancer mortality				
0–3	1	Ref.		
4–5	0.96	0.77	1.20	0.70
6	0.87	0.69	1.11	0.27
7–8	0.94	0.74	1.19	0.60
CVD mortality				
0–3	1	Ref.		
4–5	1.00	0.81	1.24	1.00
6	0.83	0.65	1.06	0.13
7–8	0.67	0.53	0.86	0.001
Heart mortality				
0–3	1	Ref.		
4–5	0.96	0.75	1.22	0.72
6	0.75	0.57	0.97	0.03
7–8	0.65	0.49	0.85	0.002
Stroke mortality				
0–3	1	Ref.		
4–5	1.23	0.76	1.99	0.41
6	1.23	0.74	2.05	0.42
7–8	0.82	0.51	1.33	0.42
AD mortality				
0–3	1	Ref.		
4–5	1.21	0.64	2.28	0.56
6	0.69	0.33	1.44	0.32
7–8	0.70	0.37	1.32	0.27
CLRD mortality				
0–3	1	Ref.		
4–5	0.92	0.62	1.38	0.70
6	0.65	0.39	1.07	0.09
7–8	0.53	0.31	0.88	0.01
Diabetes mortality				
0–3	1	Ref.		
4–5	0.57	0.33	1.00	0.05
6	0.47	0.27	0.84	0.01
7–8	0.47	0.26	0.84	0.004

Note: AD, Alzheimer’s disease; CLRD, chronic lower respiratory diseases; CVD, cardiovascular disease; HR, hazard ratio, Ref., Reference group. All analyses were adjusted for age, sex, race, education level, income level, employment status, social support, BMI, smoking status, alcohol intake, history of hypertension, history of diabetes, history of heart disease, history of stroke, and history of cancer.

**Table 3 ijerph-16-01498-t003:** Hazard ratios for all-cause, cardiovascular, and cause-specific mortality by received social integration score and social support.

Outcome	Social Integration Score ^a^				Social Support ^b^			
	HR	Lower	Upper	*p* Value	HR	Lower	Upper	*p* Value
All-cause mortality	0.92	0.90	0.94	<0.001	0.99	0.96	1.02	0.48
Cancer mortality	0.99	0.95	1.04	0.72	0.99	0.93	1.06	0.81
CVD mortality	0.91	0.87	0.95	<0.001	1.00	0.93	1.07	0.96
Heart mortality	0.91	0.87	0.95	<0.001	1.00	0.93	1.08	1.00
Stroke mortality	0.91	0.85	0.98	0.02	0.97	0.83	1.12	0.66
AD mortality	0.91	0.82	1.01	0.08	1.07	0.89	1.28	0.48
CLRD mortality	0.86	0.79	0.94	0.001	1.00	0.89	1.13	0.94
Diabetes mortality	0.85	0.77	0.95	0.004	1.02	0.88	1.19	0.75

Note: AD, Alzheimer’s disease; CLRD, chronic lower respiratory diseases; CVD, cardiovascular disease; HR, hazard ratio. ^a^ Social integration scores were considered as continuous variables. The reference group was the 0 score. All analyses were adjusted for age, sex, race, education level, income level, employment status, social support, BMI, smoking status, alcohol intake, history of hypertension, history of diabetes, history of heart disease, history of stroke, and history of cancer. ^b^ Social support was considered as a continuous variable. The reference group was the combination of rarely and never categories. All analyses adjusted for age, sex, race, education level, income level, employment status, social integration index, BMI, smoking status, alcohol intake, history of hypertension, history of diabetes, history of heart disease, history of stroke, and history of cancer.

**Table 4 ijerph-16-01498-t004:** Hazard ratios for all-cause, cardiovascular, and cause-specific mortality by received social support.

Social Support	HR	Lower	Upper	*p* Value
All-cause mortality				
Never/Rarely	1	Ref.		
Sometimes	1.02	0.88	1.19	0.79
Usually	0.99	0.87	1.13	0.87
Always	1.01	0.89	1.15	0.85
Cancer mortality				
Never/Rarely	1	Ref.		
Sometimes	0.99	0.74	1.32	0.95
Usually	0.94	0.72	1.23	0.66
Always	1.00	0.77	1.29	0.98
CVD mortality				
Never/Rarely	1	Ref.		
Sometimes	1.04	0.77	1.42	0.79
Usually	0.95	0.72	1.25	0.72
Always	1.00	0.77	1.31	0.99
Heart mortality				
Never/Rarely	1	Ref.		
Sometimes	1.02	0.75	1.40	0.89
Usually	0.93	0.68	1.26	0.64
Always	0.98	0.73	1.33	0.92
Stroke mortality				
Never/Rarely	1	Ref.		
Sometimes	0.88	0.48	1.61	0.67
Usually	0.98	0.58	1.67	0.95
Always	1.03	0.60	1.74	0.92
AD mortality				
Never/Rarely	1	Ref.		
Sometimes	1.21	0.64	2.28	0.56
Usually	0.69	0.33	1.44	0.32
Always	0.70	0.37	1.32	0.27
CLRD mortality				
Never/Rarely	1	Ref.		
Sometimes	1.22	0.67	2.22	0.51
Usually	1.18	0.68	2.04	0.55
Always	1.04	0.61	1.77	0.89
Diabetes mortality				
Never/Rarely	1	Ref.		
Sometimes	1.95	0.82	4.60	0.13
Usually	2.03	0.90	4.55	0.09
Always	1.33	0.57	3.09	0.51

Note: AD, Alzheimer’s disease; CLRD, chronic lower respiratory diseases; CVD, cardiovascular disease; HR, hazard ratio, Ref., Reference group. All analyses were adjusted for age, sex, race, education level, income level, employment status, social integration index, BMI, smoking status, alcohol intake, history of hypertension, history of diabetes, history of heart disease, history of stroke, and history of cancer.

**Table 5 ijerph-16-01498-t005:** Stratified analysis of all-cause mortality risk by social integration index.

Subgroup	Social Integration Index												
	I	II				III				IV			
Age		HR	Lower	Upper	P value	HR	Lower	Upper	P value	HR	Lower	Upper	P value
<45	Ref.	0.85	0.60	1.20	0.36	0.93	0.66	1.30	0.68	0.78	0.54	1.12	0.18
45–65	Ref.	0.86	0.71	1.04	0.12	0.65	0.51	0.81	<0.001	0.67	0.54	0.83	<0.001
>65	Ref.	1.01	0.88	1.17	0.86	0.81	0.70	0.95	<0.001	0.75	0.65	0.87	<0.001
Sex													
Women	Ref.	1.00	0.85	1.17	0.95	0.81	0.69	0.97	0.02	0.72	0.61	0.85	<0.001
Men	Ref.	0.89	0.78	1.02	0.10	0.73	0.61	0.86	<0.001	0.70	0.60	0.82	<0.001
Education													
Low	Ref.	0.99	0.84	1.16	0.85	0.88	0.74	1.06	0.17	0.81	0.69	0.95	0.01
Moderate	Ref.	0.83	0.68	1.02	0.08	0.63	0.50	0.79	<0.001	0.63	0.51	0.77	<0.001
High	Ref.	0.98	0.81	1.18	0.80	0.80	0.64	0.99	0.04	0.68	0.55	0.84	<0.001
Income													
Low	Ref.	1.12	0.90	1.38	0.30	0.84	0.65	1.08	0.18	0.86	0.67	1.10	0.23
Moderate	Ref.	0.91	0.79	1.05	0.18	0.75	0.65	0.88	0.00	0.68	0.58	0.78	<0.001
High	Ref.	0.87	0.66	1.14	0.31	0.73	0.55	0.97	0.03	0.68	0.51	0.89	0.01
Employment status													
Employed	Ref.	0.91	0.74	1.13	0.39	0.85	0.68	1.07	0.16	0.73	0.58	0.92	0.01
Retired	Ref.	0.97	0.84	1.13	0.73	0.79	0.67	0.92	0.004	0.74	0.64	0.85	<0.001
Not currently working	Ref.	0.99	0.77	1.28	0.96	0.73	0.53	0.99	0.04	0.70	0.52	0.95	0.02
Has never worked	Ref.	0.73	0.47	1.12	0.14	0.77	0.47	1.27	0.30	0.75	0.48	1.15	0.19

Note: HR, hazard ratio; Ref., Reference group. All analyses adjusted for age, sex, race, education level, income level, employment status, social support, BMI, smoking status, alcohol intake, history of hypertension, history of diabetes, history of heart disease, history of stroke, and history of cancer.

**Table 6 ijerph-16-01498-t006:** Sensitivity analysis for hazard ratios for all-cause, cardiovascular, and cancer mortality by received social integration score.

	HR	Lower	Upper	*p* Value
All-cause mortality				
Participants who died >2 years after the interview	0.89	0.87	0.91	<0.001
Participants free of cancer and CVD	0.89	0.86	0.91	<0.001
Participants with cancer	0.87	0.83	0.91	<0.001
Participants with CVD	0.89	0.86	0.93	<0.001
Cancer mortality				
Participants who died >2 years after the interview	0.97	0.93	1.02	0.197
Participants free of cancer and CVD	0.96	0.91	1.01	0.088
Participants with cancer	0.97	0.89	1.06	0.466
Participants with CVD	1.04	0.94	1.16	0.444
CVD mortality				
Participants who died >2 years after the interview	0.86	0.82	0.90	<0.001
Participants free of cancer and CVD	0.87	0.82	0.91	<0.001
Participants with cancer	0.82	0.73	0.92	0.001
Participants with CVD	0.85	0.79	0.92	<0.001

Note: CVD, cardiovascular disease; HR, hazard ratio. The social integration scores were considered as continuous variables. The reference group was 0 score. All analyses were adjusted for age, sex, race, education level, income level, employment status, social support, BMI, smoking status, alcohol intake, history of hypertension, history of diabetes, history of heart disease, history of stroke, and history of cancer.

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
