# Peer review of "Social Integration, Social Support, and All-Cause, Cardiovascular Disease and Cause-Specific Mortality: A Prospective Cohort Study"

_ijerph, 2019, doi:10.3390/ijerph16091498_

Round 1
Reviewer 1 Report
p. 2, lines 50-54: please state objectives and sub-objectives as separate sentences. The long, long sentence requires parsing to comprehend exactly what the authors propose as their objectives. At the end of the study it becomes clear that there are actually three objectives--assess social relationships, assess social support, and assess the differences between social relationships and social support, as they relate to the health outcomes selected. Since the design is prospective cohort, make it clear in the statement of objectives what followup will be done and how it will be measured.
This is a complex study, apparently well done, but difficult to read because the objectives and methods are not clearly structured at the introduction.
p.3 Line 110: statistical significance level of p < 0.05 seems a low bar for such a large sample.
Table 2--the title is confusing. The data shown look like this could be two separate tables. Or did you somehow compare the social integration index to social support as the title indicates?
Discussion: lines 197-198: unsupported conculsion; lines 204-205: unsupported conclusion;lines 225-226: unsupported conclusion; etc.
Please do not state causative conclusions based upon associative data in a non-randomized, non-controlled study.
Author Response
Response to Reviewer 1 Comments
Dear reviewer,
Thank you for your comments concerning our manuscript. These comments are valuable and very helpful for revising and improving our paper. With regards to the suggestions provided for our paper, the full text has been carefully revised. The revised sections of the manuscript are marked in blue. Our responses to your comments are shown below.
Point 1:p. 2, lines 50-54: please state objectives and sub-objectives as separate sentences. The long, long sentence requires parsing to comprehend exactly what the authors propose as their objectives. At the end of the study it becomes clear that there are actually three objectives--assess social relationships, assess social support, and assess the differences between social relationships and social support, as they relate to the health outcomes selected. Since the design is prospective cohort, make it clear in the statement of objectives what followup will be done and how it will be measured. This is a complex study, apparently well done, but difficult to read because the objectives and methods are not clearly structured at the introduction.
Response 1: Thank you for your suggestion. In the updated manuscript, we have stated our objectives and sub-objectives as separate sentences. The baseline data for this study were retrieved from the 2001 National Health Interview Survey which were linked to the National Death Index (NDI) records through December 31, 2015, using a probabilistic matching algorithm to determine mortality status. All NHIS participants ≥ 18 years of age were eligible for mortality follow-up. Participants not matched with a death record were considered alive during the follow-up period. The prospective study is with a 15-year follow-up. We revised the sections of objectives and methods at the introduction and make it read clear. These relevant sections have been revised in the updated version.
Point 2:p.3 Line 110: statistical significance level of p < 0.05 seems a low bar for such a large sample.
Response 2: Although the sample was large, the number of the events after follow-up were not large. During 0.4 million person-years of follow-up, there were only 252 deaths from stroke, 255 from CLRD, 175 from diabetes, and 128 from Alzheimer’s disease. Therefore, statistical significance was set at P < 0.05.
Point 3:Table 2--the title is confusing. The data shown look like this could be two separate tables. Or did you somehow compare the social integration index to social support as the title indicates?
Response 3: We have divided this table into two tables in the updated version. Table 2 showed the risk of all-cause, cardiovascular, and cause-specific mortality by received social integration index. Table 4 showed the risk of all-cause, cardiovascular, and cause-specific mortality by received social support.
Point 4:Discussion: lines 197-198: unsupported conclusion; lines 204-205: unsupported conclusion; lines 225-226: unsupported conclusion; etc.
Response 4: Regarding your suggestion. The relevant sections have been revised in the updated version.
Point 5:Please do not state causative conclusions based upon associative data in a non-randomized, non-controlled study.
Response 5: Thanks for your suggestion. The statements of causative conclusions have been changed. The relevant sections have been revised in the updated version.
Thank you very much for your support. Again, thank you for considering our work. Please feel free to let me know if you have any questions. Best wishes for you.
Reviewer 2 Report
This is a research of high quality of sampling and statistical procedures employed but lacking stronger input and implications in public health and incomplete methodology, so it needs to be improved.
Section 2.2 should involve detailed procedures description and information about follow-up. Information about year of baseline characteristics collection and study entry is missing and duration of follow-up is not presented. Could this section be renamed into more informative? ("Mortality surveillance and follow-up" or "Assessment of mortality events and follow-up").
In the section 2.3 the authors provide some references but information about any validation procedures of the main predictors employed (social integration and social support) is missing. In addition, authors need to justify their decision to classify Social Integration Index into 4 groups. Based on what? Why score 0 as indicating no integration at all wasn't set as reference?
Section 2.4 provides no information about covariates. Methodology with the references should be provided about assessment of BMI (was it self-reported or objectively measured), lifestyle habits (smoking, alcohol intake) and history of the disease.
Rows 102-105 in the statistical analysis section should be revised: "Exclusion of individuals who died within..." - please finish, "Calculating mortality hazards of marital status and remainder of the measure items..." - unclear, please rework.
Table 2 should fit in one page. It is complicated to read it in two pages. All titles of the tables (2-5) should be focused on mortality risk not just associations. Title of Table 1 should provide more information about the source of baseline characteristics presented.
The order of comments and tables in inappropriate and very difficult to follow, and should be revised carefully. Comments in the results section are excessive and repeating tables' information.
Title of the table 4 should be reconsidered into "Stratified analysis of all-cause mortality risk by social integration index". Abbreviations after table 4 are excessive (CVD), may be copy-pasted.
Together with tables 2-5 in the footnotes information about covariates and adjustments made should appear.
Tables 3 and 5 doesn't indicate reference group or any explanation how social integration index was employed to predict mortality risk is missing.
Finally, could the authors together with acknowledgements (or in additional section) explain their collaboration background with US CDC NHIS?
Author Response
Response to Reviewer 2 Comments
Dear reviewer,
Thank you for your comments concerning our manuscript. These comments are valuable and very helpful for revising and improving our paper. With regards to the suggestions provided for our paper, the full text has been carefully revised. The revised sections of the manuscript are marked in blue. Our responses to your comments are shown below.
Point 1:Section 2.2 should involve detailed procedures description and information about follow-up. Information about year of baseline characteristics collection and study entry is missing and duration of follow-up is not presented. Could this section be renamed into more informative? ("Mortality surveillance and follow-up" or "Assessment of mortality events and follow-up").
Response 1: Thanks for your suggestion. In 2001, 33,326 sample adults completed the NHIS. The survey records were matched to the NDI and subsequent vital status ascertainment through December 31, 2015. Person-years of follow-up were calculated for each participant from the data of the starting point to the date of death or end of the study period (31 December 2015). This section was renamed into “Assessment of mortality events and follow-up”.
Point 2: In the section 2.3 the authors provide some references but information about any validation procedures of the main predictors employed (social integration and social support) is missing. In addition, authors need to justify their decision to classify Social Integration Index into 4 groups. Based on what? Why score 0 as indicating no integration at all wasn't set as reference?
Response 2: Thanks for your suggestion. We have added the details of assignments of social integration and social support in this section 2.3. Owing to low frequencies across subgroups (age, sex, education level, income level and employment status) and low frequencies across number of cause-specific deaths, the social integration score was unavailable to analysis separately. To produce more stable estimates, social integration scores were collapsed into 4 categories of social integration index which reflected 0-3, 4-5, 6 and 7-8 social contacts.
Point 3:Section 2.4 provides no information about covariates. Methodology with the references should be provided about assessment of BMI (was it self-reported or objectively measured), lifestyle habits (smoking, alcohol intake) and history of the disease.
Response 3: Thank you for your suggestion. Demographic variables included age, sex, race (non-Hispanic white, non-Hispanic black, Hispanic, other non-Hispanic), education level (less than high school degree, high school degree or equivalent and more than high school degree), income level (family income to poverty ratio (PIR) ≤ 1, 1< PIR ≤ 4, PIR ≥4), and employment status (never worked, working, retired, or out of work). Lifestyle variables were from self-reported responses for these questionnaires. Lifestyle variables included body mass index (BMI), smoking status, alcohol intake, history of hypertension (no versus yes), history of diabetes (no versus yes), history of heart disease (no versus yes), history of stroke (no versus yes), and history of cancer (no versus yes). BMI was calculated as weight in kilograms divided by height squared and was categorized as <25, 25-30 and >30 kg/m2. Respondents were defined as never smokers if they had smoked less than 100 cigarettes in their lifetime. Former smokers are defined as those who had smoked at least 100 cigarettes in their lifetime but did not currently smoke. Participants were classified as current smokers who reported having smoked more than 100 cigarettes and currently smoked. Respondents were categorized into 3 alcohol consumption groups: 1) lifetime abstainers: <12 drinks in one’s lifetime; 2) former drinkers: ≥12 drinks in a previous year; 3) current drinker: drinking now and ≥12 drinks in one’s lifetime. This text has been revised in the updated version.
Point 4:Rows 102-105 in the statistical analysis section should be revised: "Exclusion of individuals who died within..." - please finish, "Calculating mortality hazards of marital status and remainder of the measure items..." - unclear, please rework.
Response 4: Thank you for your suggestion. The several sensitivity analyses were also conducted to evaluate the robustness of results: 1, exclusion of individuals who died within 2 years after the interview; 2, exclusion of participants with chronic diseases included heart disease, stroke and cancer at baseline; 3, limiting analyses to CVD or cancer patients. This text has been revised in the updated version.
Point 5:Table 2 should fit in one page. It is complicated to read it in two pages. All titles of the tables (2-5) should be focused on mortality risk not just associations. Title of Table 1 should provide more information about the source of baseline characteristics presented.
Response 5: Thank you for your suggestion. Table 2 have been fit in one page. All titles of the tables have been all revised in the updated version.
Point 6:The order of comments and tables in inappropriate and very difficult to follow, and should be revised carefully. Comments in the results section are excessive and repeating tables' information.
Response 6: Thanks for your suggestion. We revised the order of comments and tables in the Results section. The relevant sections have been revised in the updated version.
Point 7:Title of the table 4 should be reconsidered into "Stratified analysis of all-cause mortality risk by social integration index". Abbreviations after table 4 are excessive (CVD), may be copy-pasted.
Response 7: Thanks for your suggestion. Title of the table 4 has been changed to "Stratified analysis of all-cause mortality risk by social integration index". We have deleted the Abbreviations of CVD.
Point 8:Together with tables 2-5 in the footnotes information about covariates and adjustments made should appear.
Response 8: Thanks for your suggestion. We have added the adjustment information in the footnotes. This text has been revised in the updated version.
Point 9:Tables 3 and 5 doesn't indicate reference group or any explanation how social integration index was employed to predict mortality risk is missing.
Response 9: In the updated version, social integration score and social support were considered as continuous variable in table 3 and table 6. For social integration score, the reference group was 0 score. For social support, the reference group was the combination of rarely and never categories. In table 5, the social integration index was considered as categorical variable, the reference group was Level I of social integration index. In the updated version, we have added the relevant information in the footnotes.
Point 10:Finally, could the authors together with acknowledgements (or in additional section) explain their collaboration background with US CDC NHIS?
Response 10: The data are publicly available. The authors thank the National Center for Health Statistics of the Centers for Disease Control and Prevention for sharing the NHIS data.
Thank you very much for your support. Again, thank you for considering our work. Please feel free to let me know if you have any questions. Best wishes for you.
Round 2
Reviewer 2 Report
The paper was improved significantly, lacking information was added. Recommend to accept.